# Targeted Genomic Profiling and Chemotherapy Outcomes in Grade 3 Gastro-Entero-Pancreatic Neuroendocrine Tumors (G3 GEP-NET)

**DOI:** 10.3390/diagnostics13091595

**Published:** 2023-04-29

**Authors:** Giuseppe Lamberti, Natalie Prinzi, Alberto Bongiovanni, Mariangela Torniai, Elisa Andrini, Dario de Biase, Deborah Malvi, Mirta Mosca, Rossana Berardi, Toni Ibrahim, Sara Pusceddu, Davide Campana

**Affiliations:** 1Department of Medical or Surgical Sciences, University of Bologna, 40126 Bologna, Italy; giuseppe.lamberti8@unibo.it (G.L.); elisa.andrini3@unibo.it (E.A.); mirtamosca92@gmail.com (M.M.); 2Medical Oncology Department, IRCCS Azienda Ospedaliero-Universitaria Sant’Orsola-Malpighi di Bologna, 40138 Bologna, Italy; 3Medical Oncology, Foundation IRCCS National Cancer Institute, 20133 Milano, Italy; natalie.prinzi@istitutotumori.mi.it (N.P.); sara.pusceddu@istitutotumori.mi.it (S.P.); 4Osteoncology and Rare Tumor Center (CDO-TR), IRCCS Istituto Romagnolo per lo Studio dei Tumori (IRST) “Dino Amadori”, 47014 Meldola, Italy; alberto.bongiovanni@irst.emr.it (A.B.); toni.ibrahim@ior.it (T.I.); 5Department of Oncology, Università Politecnica delle Marche-AOU delle Marche, 60126 Ancona, Italy; mariangelatorniai@hotmail.it (M.T.); rossana.berardi@ospedaliriuniti.marche.it (R.B.); 6Department of Pharmacy and Biotechnology, University of Bologna, 40126 Bologna, Italy; dario.debiase@unibo.it; 7Solid Tumor Molecular Pathology Laboratory, IRCCS Azienda Ospedaliero-Universitaria di Bologna, 40138 Bologna, Italy; 8Pathology Unit, IRCCS Azienda-Ospedaliero Universitaria di Bologna, 40138 Bologna, Italy; deborah.malvi@aosp.bo.it

**Keywords:** NET, G3, FOLFOX, XELOX, CAPTEM, NGS, genomic, DAXX, ATRX

## Abstract

Background: Grade 3 gastro-entero-pancreatic neuroendocrine tumors (G3 GEP-NET) are poorly characterized in terms of molecular features and response to treatments. Methods: Patients with G3 GEP-NET were included if they received capecitabine and temozolomide (CAPTEM) or oxaliplatin with either 5-fluorouracile (FOLFOX) or capecitabine (XELOX) as first-line treatment (chemotherapy cohort). G3 NET which successfully undergone next-generation sequencing (NGS) were included in the NGS cohort. Results: In total, 49 patients were included in the chemotherapy cohort: 15 received CAPTEM and 34 received FOLFOX/XELOX. Objective response rate (ORR), progression-free survival (PFS), and overall survival (OS) were 42.9%, 9.0 months, and 33.6 months, respectively. Calculating a Ki67 cutoff using ROC curve analysis, tumors with Ki67 ≥ 40% had lower ORR (51.2% vs. 0%; *p* = 0.007) and shorter PFS (10.6 months vs. 4.4 months; *p* < 0.001) and OS (49.4 months vs. 10.0 months; *p* = 0.023). In patients who received FOLFOX/XELOX as a first-line treatment, ORR, PFS, and OS were 38.2%, 7.9 months, and 30.0 months, respectively. In the NGS cohort (N = 13), the most mutated genes were *DAXX*/*ATRX* (N = 5, 38%), *MEN1* (N = 4, 31%), *TP53* (N = 4, 31%), *AKT1* (N = 2, 15%), and *PIK3CA* (N = 1, 8%). Conclusions: FOLFOX/XELOX chemotherapy is active as the first-line treatment of patients with G3 GEP-NET. The mutational landscape of G3 NET is more similar to well-differentiated NETs than NECs.

## 1. Introduction

Neuroendocrine neoplasms (NENs) are rare tumors whose incidence is, however, rising [1]. NEN is a comprehensive term used to indicate both well-differentiated neuroendocrine tumors (NET), the most common being of gastro-entero-pancreatic (GEP-NET) origin, and poorly differentiated neuroendocrine carcinomas (NECs) [2]. NETs are characterized by well-differentiated neuroendocrine morphology (“organoid” cell arrangement and finely granular cytoplasm), uniform nuclei with “salt and pepper” chromatin, and express neuroendocrine biomarkers such as chromogranin A, synaptophysin, CD56/NCAM, and INSM-1. On the other hand, NECs have poorly differentiated morphology resembling small-cell or large-cell carcinomas, with sheet-like architecture, abundant necrosis, irregular nuclei, and less cytoplasmic granularity, which is reflected by decreased proportion and intensity of neuroendocrine biomarkers staining. Before 2017, the classification of a NEN in the NET or NEC group was primarily based on the proliferation index as assessed by Ki67: NET included grade 1 (G1) tumors with Ki67 < 3% and G2 tumors with Ki67 between 3 and 20%, whereas all NEN with Ki67 > 20% were G3 and classified as NEC.

NET and NEC have profoundly different clinical behavior and treatment susceptibility. Indeed, capecitabine and temozolomide (CAPTEM) or oxaliplatin and either 5-fluorouracil (FOLFOX) or capecitabine (XELOX) are the most frequently used chemotherapy in the treatment of NET, although only in select cases or after the failure of other treatments [3], whereas chemotherapy with platinum and etoposide is the standard treatment for NEC [4]. Nevertheless, the NORDIC NEC study demonstrated that response to platinum-based chemotherapy was different among NECs of GEP origin, depending on Ki67: those tumors with a Ki67 < 55% had lower response rate but longer survival outcomes than tumors with a Ki67 ≥ 55% [5]. Since 2017 for pancreatic NEN and 2019 for NEN of other sites, the G3 NET subgroup has been introduced by World Health Organization (WHO) classification, which indicates NEN with a well-differentiated morphology (NET) but with a Ki67 > 20%, which is, however, usually at <50–70% [2].

To date, this newly defined G3 NET category has been poorly characterized in terms of clinicopathological and genomic features and response to treatment. We sought to characterize the outcomes of patients with G3 GEP-NET to first-line chemotherapy and to describe their mutational landscape.

## 2. Materials and Methods

### 2.1. Patient Selection

Consecutive patients with histologically proven G3 NET have been included in the study if they have received chemotherapy with either FOLFOX (oxaliplatin and 5-fluorouracile), XELOX (oxaliplatin and capecitabine), or CAPTEM (capecitabine and temozolomide) as first-line treatment of a locally advanced or metastatic GEP-NET (*chemotherapy cohort*), or if their tumor had successfully undergone targeted next-generation sequencing (*NGS cohort*).

G3 NETs were defined as NET with well-differentiated morphology and proliferation index ≥ 20% as assessed by Ki67 (Mib-1 clone) or ≥20 mitosis per 10 high-powered fields, according to WHO 2019 classification after revision by a NEN-dedicated pathologist. NECs with poorly differentiated morphology (either small cell or large cell) and NET with Ki67 ≤ 20% were excluded.

In the chemotherapy cohort, clinicopathological features have been correlated with chemotherapy outcomes, namely objective response rate (ORR), progression-free survival (PFS), and overall survival (OS). Response to treatment was assessed according to RECIST v1.1 criteria [6].

This study was approved by local IRB (Comitato Etico Indipendente, IRCCS Policlinico Sant’Orsola-Malpighi of Bologna, protocol code: SOCRATE, 787/2019/Oss/AOUBo) and was conducted in accordance with the principles of the Declaration of Helsinki (revision of Edinburgh, 2000).

### 2.2. Next-Generation Sequencing

The next-generation sequencing (NGS) analysis has been performed using two lab-developed panels of genes developed in the Molecular Pathology Laboratory of Sant’Orsola-Malpighi Hospital. These two panels allow the analysis of the entire coding regions (CDS) or hot-spot regions of 21 genes, for a total of 937 amplicons (about 82.04 kb), starting from archived formalin-fixed and paraffin-embedded (FFPE) tissues. The entire CDS of *ATRX*, *DAXX*, *EP300*, *MEN1*, *EIF1AX*, *NOTCH1*, *NOTCH2*, *NOTCH3*, *NOTCH4*, *PIK3CA*, *RB1*, *TP53*, *STK11*, *KEAP1*, and *NFE2L2*, or the hot-spot regions of *AKT1*, *mTOR*, *KRAS*, *NRAS*, *HRAS*, and *BRAF* have been analyzed. NGS was performed using the Gene Studio S5 sequencer (ThermoFisher Scientific Inc., Waltham, MA, USA). Briefly, about 30 ng of DNA have been used per each panel for the amplicon library preparation, performed with the AmpliSeq Plus Library Kit 2.0. Templates were prepared with Chef Machine Template and then sequenced using an Ion 530 chip, and sequences were analyzed with IonReporter 5.16 (ThermoFisher Scientific) and IGV (Integrative Genomic Viewer) tool. Mutation pathogenicity has been assessed by Varsome (for the ACMG Classification, https://varsome.com/ [accessed on 31/12/2022]), OncoKB [7], and ClinVar [8] database annotations [9]. Mutations annotated as pathogenic in either ACMG Classification, OncoKB, or ClinVar, or as probably damaging by PolyPhen-2 (score ≥ 0.95) were included in the study. Included mutations and key clinicopathological features have been displayed in an OncoPrint plot.

### 2.3. Statistical Analysis

Categorical variables were reported as absolute numbers and proportions and compared by Fisher’s test or Chi-squared test, as appropriate. Continuous variables were reported as median and range and compared by Mann-Whitney test. ORR was defined as the proportion of complete responses (CR) and partial responses (PR) out of the total evaluable cases by RECIST v1.1 [6], whereas the disease control rate (DCR) was defined as the proportion of CR, PR, and stable disease (SD) out of the total evaluable cases by RECIST v1.1. PFS was defined as the time from treatment start to RECIST-defined disease progression (PD) or death from any cause, whichever occurred first, whereas OS was defined as the time from treatment start to death from any cause. Survival times were estimated using the Kaplan–Meier method, reported in months, and 95% confidence interval (CI) estimated with the Greenwood formula, and compared by the log-rank method. Risk factors were analyzed by univariate and multivariate analysis using the Cox proportional-hazards method and expressed as hazard ratios (HR) [95% CI]. The multivariate model was fitted including all variables with *p*-value < 0.1 significance in the univariate analysis. The area under the receiver-operating-characteristic (ROC) curve was evaluated to estimate the best Ki67 cut-off to predict response. The best cut-off value was estimated by using Youden’s statistics [10]. The *p*-values were considered significant when <0.05. The statistical analysis was carried out using R Statistical package version 4.2.2 software.

## 3. Results

Overall, 52 patients have been included in the study, of which N = 49 received either CAPTEM or FOLFOX/XELOX as first-line treatment for their G3 NET (“Chemotherapy cohort”) and N = 13 had available tissue to perform NGS (Figure 1).

### 3.1. Chemotherapy Cohort

Characteristics of the patients in the chemotherapy cohort are summarized in Table 1.

Median age was 60 years (range 18–80), primary site was pancreas (panNET) in 32 (65%) and the gastrointestinal tract in 17 (35%), median Ki67 was 30% (range 20–50%) and stage at the time of chemotherapy was metastatic in all tumors except 4 (8%), which were locally advanced. In those with available pre-treatment nuclear medicine imaging, the tumor was positive at the 18F-FDG PET/CT scan in 23 cases (89%) and negative in 3 cases (11%), whereas it was positive at the 68Ga-DOTANOC-PET/CT scan in 30 (75%) and negative in 10 (25%). Patient characteristics were overall well balanced between the FOLFOX/XELOX group (N = 34) and the CAPTEM group (N = 15), except for the higher rate of 18-F-FDG-PET positivity in the FOLOX/XELOX group as compared to the CAPTEM group (100% [N = 16/16] vs. 70% [N = 7/10], respectively; *p* = 0.046).

CR was observed in one case (2.0%), PR in 20 (40.8%), and SD in 17 (34.7%), with an ORR of 42.9% (95%CI: 28.8–57.8, N = 21/49, Appendix A) and a DCR of 77.6% (95%CI: 63.4–88.2, N = 38/49). Progressive disease (PD) was the best response in 11 cases (22.4%). Ki67 tended to be lower in responders as compared to non-responders (median 25% vs. 30%, respectively; *p* = 0.094) (Figure 2).

At a median follow-up of 24.8 months (95%CI: 20.2–48.6), PFS and OS were 9.0 months (95%CI: 7.2–13.0), and 33.6 months (95% CI: 19.2–NA) (Appendix A). PFS was shorter in patients with FDG-avid tumors than in those with tumors with no uptake at the 18F-FDG PET/CT scan (8.1 months [95%CI: 7.0–26.5] vs. 41.8 months [95%CI: 41.8–NA], respectively; *p* = 0.01) (Appendix A) and tended to be shorter in patients who received FOLFOX/XELOX than in those who received CAPTEM (7.9 months [95%CI: 6.8–11.4] vs. 13.0 months [95%CI: 7.0–NA], respectively; *p* = 0.098). Nevertheless, after adjusting for potential confounding factors in a multivariable Cox proportional-hazards model, only Ki67 was independently associated with the risk of progression or death (HR: 1-07 [95%CI: 1.02–1.13]; *p* = 0.01) (Appendix A). Indeed, after excluding the three patients with no uptake at the 18F-FDG PET imaging who have all received CAPTEM treatment, we observed similar outcomes in terms of ORR (38% vs. 50%, *p* = 0.514), PFS (7.9 months [95%CI: 6.8–11.4] vs. 11.7 months [95%: 6.2–NA]; *p* = 0.6), and OS (30.0 months [95%CI: 13.8–NA] vs. 19.2 months [95%CI: 15.0–NA]; *p* = 0.7) in the FOLFOX/XELOX and CAPTEM group, respectively, further supporting the absence of significant difference between the two treatments (Appendix A).

By means of ROC curve analysis, we then estimated 40% to be the best Ki67 cutoff to predict response, with 100% sensitivity, 29% specificity, and 0.640 AUC (Figure 3A). When comparing with the group with Ki67 < 40%, the group with tumors with Ki67 ≥ 40% had lower ORR (51.2% [95%CI: 35.1–67.1] vs. 0% [95%CI: 0–31.1]; *p* = 0.007) and shorter PFS (10.6 months [95%CI: 8.1–24.8] vs. 4.4 months [95%CI: 2.2–NA]; *p* < 0.001) and OS (49.4 months [95%CI: 25.5–NA] vs. 10.0 months [95%CI: 6.0–NA]; *p* = 0.023) (Figure 3B–D).

We then focused on outcomes to FOLFOX/XELOX because the activity of these chemotherapy schedules has been incompletely characterized, especially in G3 GEP-NET. In the subgroup of patients who received FOLFOX/XELOX as first-line treatment, ORR, PFS, and OS were 38.2% (95%CI: 22.2–56.4), 7.9 months (95%CI: 6.8–11.4), and 30.0 months (95% CI: 13.8–NA), respectively (Figure 4).

While there was no difference in Ki67 between responders and non-responders to FOLFOX/XELOX chemotherapy (median 28% vs. 30%, respectively; *p* = 0.461), increasing Ki67 was associated with both shorter PFS (HR: 1.09 [95%CI: 1.03–1.16]; *p* = 0.006) and OS (HR: 1.11 [95%CI: 1.03–1.20]; *p* = 0.004). We then estimated the best cutoff to predict response in the FOLFOX/XELOX cohort as well by means of ROC curve analysis, which was 40% as in the overall chemotherapy cohort (Appendix A). When comparing with the group with Ki67 <40%, the Ki67 ≥ 40% group had numerically lower ORR (46.4% [95%CI: 27.5–66.1] vs. 0% [95%CI: 0–45.9]; *p* = 0.062) and significantly shorter PFS (9.4 months [95%CI: 7.4–24.8] vs. 2.4 months [95%CI: 1.0–NA]; *p* < 0.001) and OS (49.5 months [95%CI: 25.5–NA] vs. 7.5 months [95%CI: 6.0–NA]; *p* < 0.001) (Appendix A).

Because outcomes to CAPTEM of patients with panNET have been extensively described [11], we analyzed outcomes of the N = 22 patients with G3 panNET who received FOLFOX/XELOX as first-line chemotherapy: in this cohort, ORR was 36.4%, median PFS was 7.9 months (95%CI: 5.0–32.6), and median OS was 49.4 months (95%CI: 9.8–NA) (Appendix A).

### 3.2. NGS Cohort

The NGS cohort only partially overlaps with the chemotherapy cohort and comprises 13 patients with G3 NETs whose tumors had successfully undergone targeted NGS with a custom panel. The patient characteristics are summarized in Table 2. The median age in this cohort was 53 years (range: 18–83), and 7 (54%) were men. Most of the patients had a pancreatic primary (N = 10/13, 77%) with a median Ki67 of 26.5% (range: 21–35).

As shown in Figure 5, a mutation in at least one of the genes has been found in nine out of thirteen samples (69%). The most frequently mutated genes are *DAXX/ATRX* (N = 5/13, 38%), followed by *MEN1* (N = 4/13, 31%), and *TP53* (N = 4/13, 31%). Genes encoding proteins of the Pi3k/Akt/mTor pathway are also mutated in 3/13 (23%) of tumors: *AKT1* in two cases (15%) and *PIK3CA* in one case (8%). Interestingly, two pancreatic NETs harbored a mutation in genes of the RAS-family, namely *KRAS* (with *TP53* co-mutation) and *HRAS* (with *AKT1* and *DAXX* co-mutation). Notably, no mutation in *RB1* has been identified. In this cohort, six patients received FOLFOX/XELOX chemotherapy, of whom five achieved a PR, whereas eight patients received CAPTEM, of whom four achieved a partial response. One patient with a pancreatic G3 NET received both treatment regimens and responded to both. Given the small sample size, no meaningful comparison can be made according to the treatment or mutation profile.

## 4. Discussion

We have shown that FOLFOX/XELOX chemotherapy is active in G3 GEP-NET, especially in tumors with Ki67 < 40% and that uptake at 18F-FDG PET is prognostic in G3 panNETs. Furthermore, G3 NETs demonstrated a genomic profiling similar to those of well-differentiated NETs, with a high rate of mutations in *DAXX*, *ATRX*, and *MEN1*, and less similar to poorly differentiated NECs, given the absence of mutations in *RB1* and in oncogenes such as *KRAS* and *BRAF*.

As the G3 NET category has been recently defined, it poses unique challenges in its management as it shares clinical-pathologic features with both well-differentiated NET and with poorly differentiated NEC, which have dramatically different treatment susceptibility. Indeed, platinum-etoposide chemotherapy, which is the standard treatment in advanced NEC, has shown to be less active in G3 NETs with ORR as low as 24% [12]. On the other hand, chemotherapy is seldom used in G1-2 NET, only in select cases or after failure of other treatments, such as somatostatin analogues at standard or non-conventional doses, peptide radionuclide receptor therapy (PRRT), or tyrosine kinase inhibitors (namely everolimus and sutent for panNET) [3,4,13,14,15,16,17,18,19,20]. Chemotherapy with CAPTEM or single-agent temozolomide has been extensively investigated in G1-2 NETs, especially in panNETs, and yielded ORR of 30–70% and median PFS and OS of up to 23 and over 70 months, respectively [11,21,22,23,24,25,26,27]. Less is known about the activity of temozolomide or CAPTEM in patients with G3 NET. A few small retrospective heterogenous series of temozolomide and CAPTEM in G3 NET report ORR of 44–50% and median PFS of 10 months, which compare favorably with our findings (ORR 53% and PFS 13 months in patients who received CAPTEM as first-line treatment) [28,29]. Even less data are available on the use of FOLFOX/XELOX in G3 NET. Evidence of the activity of FOLFOX/XELOX in G3 NET comes from relatively small heterogeneous retrospective series, which included few G3 NET patients (10 to 12 cases), who mostly received FOLFOX/XELOX as lines of treatment later than first, yielding ORR of 26–45% and PFS of 6–8 months [30,31,32]. In this scenario, we reported about the largest series to date of 34 patients with G3 GEP-NET who received FOLFOX/XELOX as first-line treatment and achieved an ORR of 38% and a median PFS of 7.9 months. Despite the ROC performances likely limited by the relatively small sample size, we found that a Ki67 cutoff of 40% can discriminate outcomes in G3 NET. Indeed, no objective response has been observed with neither CAPTEM nor FOLFOX/XELOX in tumors with Ki67 ≥ 40%, which demonstrated shorter PFS and OS when compared to those with Ki67 < 40%. This was similar to what was previously reported in a large heterogeneous series of patients with NEN with Ki67 > 10% treated with CAPTEM [33]. Interestingly, in patients with NEC, defined as Ki67 > 20%, who received platinum etoposide chemotherapy in the NORDIC NEC study, the ORR was lower in those with Ki67 < 55% than those with a Ki67 ≥ 55% [5]. In an attempt to reconcile these findings with those from the present study, it can be speculated that platinum-etoposide chemotherapy is more effective in G3 NET or NEC with Ki67 ≥ 40–55%, whereas other chemotherapy regimens, such as FOLFOX/XELOX or CAPTEM might be more active in tumors with lower Ki67. With the available data, it is unclear whether the two different types of chemotherapy have differential activity in NEN with Ki67 between 40% and 55%.

In the NGS cohort, a high proportion of mutations in genes which are commonly found in well-differentiated NET, such as *DAXX*, *ATRX*, and *MEN1*, as well as alterations in genes encoding proteins of the PI3K/Akt/mTOR pathway have been found [34,35,36,37,38]. The pattern of mutated genes appears to be different from those of NEC, which are characterized by co-occurring alteration in *TP53* and *RB1*, and in the driver oncogenes *KRAS* and *BRAF* [39,40], the latter especially in colorectal NEC [41]. Mutations in *TP53* have been identified also in our cohort (31% of cases), but none of these mutations co-occurred with *RB1* alterations. Overall, these findings suggest that the genomics of G3 NETs are more similar to G1–G2 NETs than NECs.

Limitations of this study include its retrospective design, which introduced a potential bias in chemotherapy assignment since patients with less aggressive tumors at local investigator’s judgement might have been more likely to receive CAPTEM than FOLFOX/XELOX, which could explain the non-significant numeric differences in outcomes between the two treatment schedules. The retrospective data collection was, however, inevitable since the G3 NET category has been recently defined and represents a rare subgroup of a low-incidence disease. Moreover, some subgroups are small in size and prevent definitive or meaningful comparisons (e.g., tumors with no uptake at 18F-FDG-PET or patients who received chemotherapy in the NGS cohort). Nevertheless, we reported the largest series to date of G3 GEP-NET patients treated with first-line chemotherapy, including FOLFOX/XELOX chemotherapy. Lastly, we performed targeted NGS of coding regions of a selected list of genes which might have missed alterations in intronic regions of suppressor oncogenes, e.g., *RB1*.

## 5. Conclusions

In conclusion, FOLFOX/XELOX chemotherapy is active as first-line treatment for patients with G3 GEP-NET and should be considered an option in this setting. The mutational landscape of G3 NET is more similar to well-differentiated NET than to NEC.

## Figures and Tables

**Figure 1 diagnostics-13-01595-f001:**
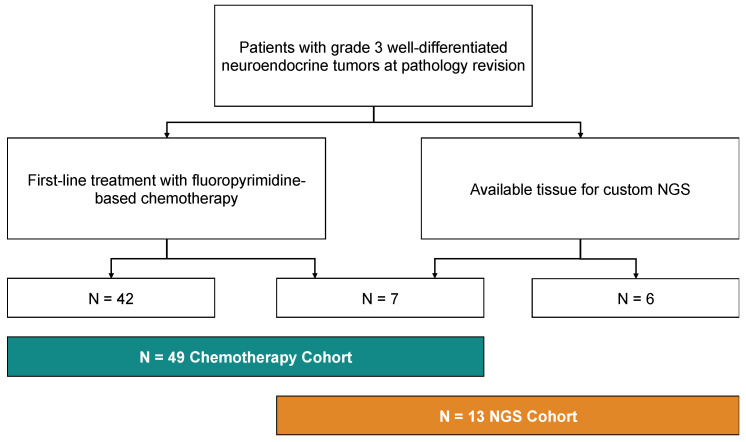
Consort diagram of the study, showing the allocation of patients in the “Chemotherapy cohort” and in the “NGS cohort”.

**Figure 2 diagnostics-13-01595-f002:**
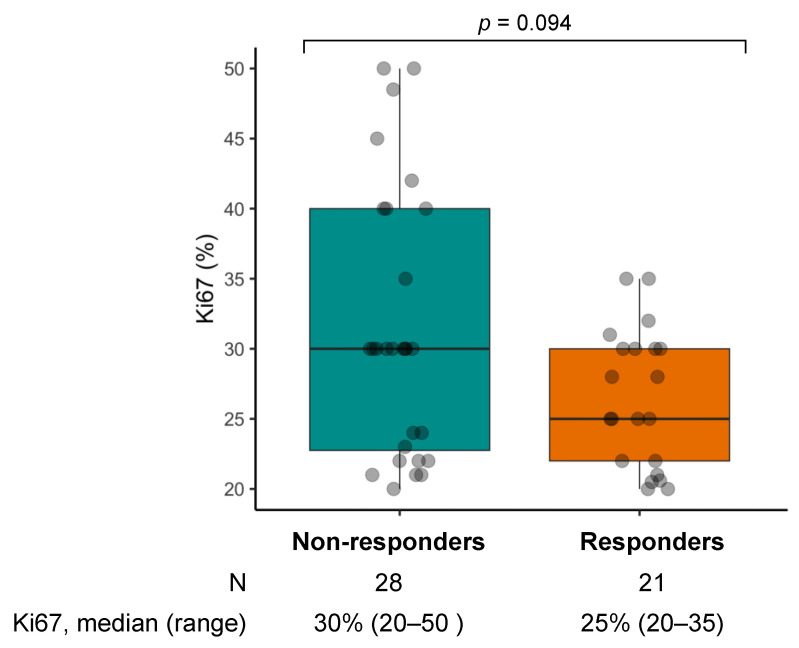
Box and whiskers plot of Ki67 by objective response in the chemotherapy cohort.

**Figure 3 diagnostics-13-01595-f003:**
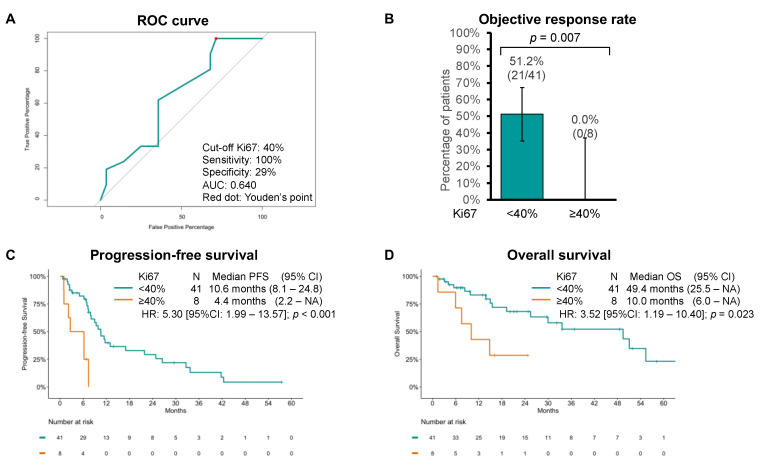
(**A**) Receiving operator characteristics (ROC) curve of objective response by Ki67 cutoffs. Youden’s point is represented by a red dot. (**B**) Objective response rate, (**C**) progression-free survival (PFS), and (**D**) overall survival (OS) estimates using the Kaplan–Meier method to chemotherapy by Ki67 cutoff of 40%.

**Figure 4 diagnostics-13-01595-f004:**
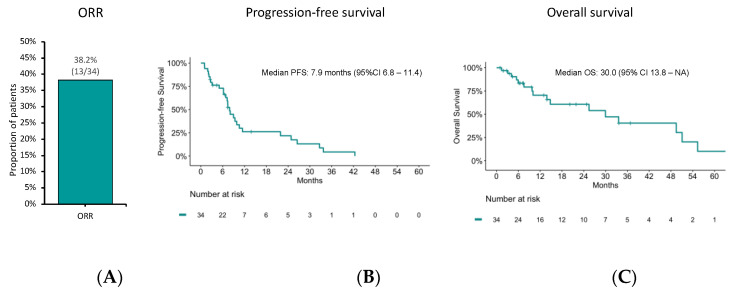
(**A**) Objective response rate (ORR), (**B**) progression-free survival (PFS), and (**C**) overall survival (OS) estimates using the Kaplan–Meier method to FOLFOX/XELOX chemotherapy.

**Figure 5 diagnostics-13-01595-f005:**
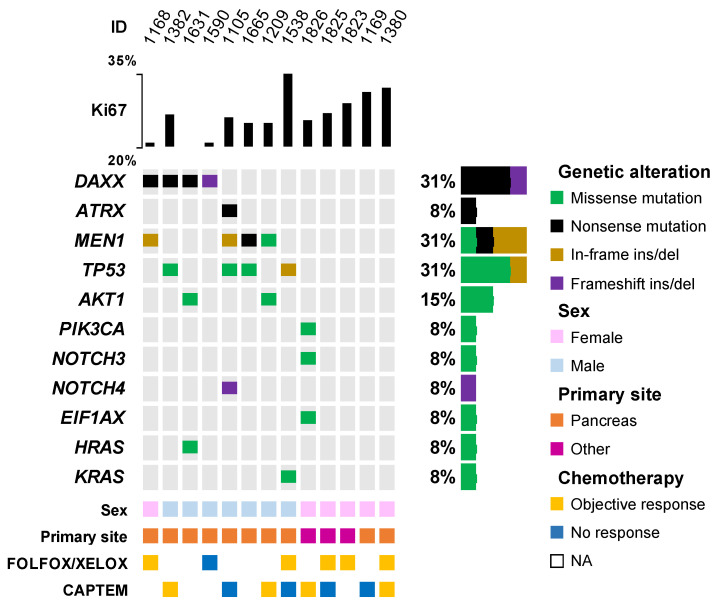
OncoPrint plot of mutations in the NGS cohort. Only genes altered in at least one sample are displayed. NA: not available. “Other” primary site category includes lung (n = 1), rectum (n = 1), and ovary (n = 1).

**Table 1 diagnostics-13-01595-t001:** Characteristics of patients in the chemotherapy cohort by treatment regimen.

	Overall (N= 49)	CAPTEM (N= 15)	FOLFOX/XELOX (N= 34)	*p*
**Age**				
Median [range]	60.0 [18.0, 80.0]	63.0 [39.0, 77.0]	58.5 [18.0, 80.0]	0.345
**Site of primary**				
Pancreas	32 (65.3%)	10 (66.7%)	22 (64.7%)	1.00
GI	17 (34.7%)	5 (33.3%)	12 (35.3%)	
**Stage**				
IIIB	4 (8.16%)	2 (13.3%)	2 (5.88%)	0.576
IV	45 (91.8%)	13 (86.7%)	32 (94.1%)	
**Ki67**				
Median [range]	30.0 [20.0, 50.0]	28.0 [20.0, 48.5]	30.0 [20.0, 50.0]	0.599
**18F-FDG-PET**				
Negative	3 (11.5%)	3 (30.0%)	0 (0%)	** *0.046* **
Positive	23 (88.5%)	7 (70.0%)	16 (100%)	
**68Ga-DOTANOC-PET**				
Negative	10 (25.0%)	3 (27.3%)	7 (24.1%)	1.00
Positive	30 (75.0%)	8 (72.7%)	22 (75.9%)	
**Concomitant use of SSA**	25 (51.0%)	9 (60.0%)	16 (47.1%)	0.538

**Table 2 diagnostics-13-01595-t002:** Characteristics of patients in the NGS cohort.

	Overall (N = 13)
**Age**	
Median [range]	53.0 [18.0, 83.0]
Sex	
Female	6 (46%)
Male	7 (54%)
**Site of primary**	
Pancreas	10 (77%)
Other *	3 (33%)
**Stage**	
IIIB	5 (38%)
IV	8 (62%)
**Ki67**	
Median [range]	26.5 [21.0, 35.0]
**18F-FDG-PET/CT**	
Negative	1 (11%)
Positive	8 (89%)
**68Ga-DOTANOC-PET/CT**	
Negative	0 (0%)
Positive	10 (100%)
**Chemotherapy**	
FOLFOX/XELOX	6 (46%)
CAPTEM	10 (69%)

* includes lung (n = 1), rectum (n = 1), and ovary (n = 1).

## Data Availability

Data supporting the present study are available from thecorresponding author upon reasonable request.

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
