# Peer review of "Targeted Genomic Profiling and Chemotherapy Outcomes in Grade 3 Gastro-Entero-Pancreatic Neuroendocrine Tumors (G3 GEP-NET)"

_diagnostics, 2023, doi:10.3390/diagnostics13091595_

Round 1
Reviewer 1 Report
Great read, thanks authors.
I am not entirelly sure about the ROC analysis, we see a very little AUC in this result, which may be due to low n number.
Would be great if this data or at least summary statistics are deposited in a well recognised data storige provider so other researchers can benifit from author efforts.
What are the Other site in Primary tissue sites - its important to disclose this as this may lead to analytic bias.
I would also appreciate more efforts in integrating data available with the other big consortium projects such as:
Would be great if authors could put the efforts in the perspective of other projects that have looked at the G3 GEP-NET: For example TCGA and Would be great if authors could look at whether same mutations are detected.
Are there any studies that include patients that undergo treatments with lanreotide, i believe this is drug currently used for this types of cancer? Perhaps authors could put their study in perspective of CLARINET PLUS.
some minor improvements can be made.
Author Response
We thank the reviewer for taking the time to review our manuscript and for the precious comments. By addressing those, we believe that the overall quality of the manuscript is sensibly increased and hope it is now suitable for publication in the Journal.
Great read, thanks authors.
I am not entirelly sure about the ROC analysis, we see a very little AUC in this result, which may be due to low n number.
We thank the reviewer for this insightful comment. We agree with the reviewer that the ROC AUC has a somehow low estimate, which we agree it is likely largely dependent on the small sample size, that is nevertheless not negligible considering the rarity of G3 NET. We commented on this in the Discussion session, lines 272-274.
Would be great if this data or at least summary statistics are deposited in a well recognised data storige provider so other researchers can benifit from author efforts.
We thank the reviewer for this suggestion. Unluckily, local privacy policy does not allow the authors to do so.
What are the Other site in Primary tissue sites - its important to disclose this as this may lead to analytic bias.
We thank the reviewer for this suggestion. The three G3 NET classified as “Other” primary were of pulmonary, rectal, and ovarian origin, one each. This piece of information has been specified in Table 2 footnote and in Figure 5 legend.
I would also appreciate more efforts in integrating data available with the other big consortium projects such as:
Would be great if authors could put the efforts in the perspective of other projects that have looked at the G3 GEP-NET: For example TCGA and Would be great if authors could look at whether same mutations are detected.
We thank the reviewer for this insightful suggestion. Unluckily, neither TCGA or GENIE include G3 neuroendocrine tumors genomic data or, more likely, the annotation for this tumor type is unreliable as it has been recently included in the WHO classification. Indeed, the largest neuroendocrine study available in cBioportal (Scarpa et al, Nature 2017, ref n. 34 in the manuscript) included 98 pancreatic NETs from several referral Institutions worldwide and only 4 were likely G3 pancreatic NETs, i.e. reported as well-differentiated and grade 3 tumors. Among these, one tumor harbored mutations in both TP53 and RB1 genes, while no mutations in our genes of interest were found in the other three samples. Because of this limitation in the available literature, we compared the mutational landscape of our NGS cohort with available data from both well-differentiated NET series and poorly-differentiated NEC series in the Discussion section, line 286-294.
Are there any studies that include patients that undergo treatments with lanreotide, i believe this is drug currently used for this types of cancer? Perhaps authors could put their study in perspective of CLARINET PLUS.
We thank the reviewer for this comment. Because G3 NET is more aggressive as compared to G1-2 NET and somatostatin receptor expression is roughly inversely proportional to proliferation index, the use of lanreotide or other somatostatin analogue is not of choice in these patients. To the best of our knowledge, we are not aware of a CLARINET PLUS study; nevertheless, the CLARINET FORTE trial of high-dose lanreotide (120 mg every 14 days) in patients with well-differentiated G1-G2 NET has demonstrated that activity of lanreotide, even at nonconventional increased doses, yielded poor outcomes in patients with NET with Ki67 >10% (PFS 5.5 and 2.8 months in midgut and pancreatic NET, respectively). This Ki67 threshold is lower of the threshold to define G3 NET (>20%), which were excluded by design from the trial. We are not incline to include these considerations in the manuscript because of the different population considered, unless the review does feel it as being compelling.

Reviewer 2 Report
Thank you for an interesting and well-written paper.
1. Could you please describe the NECs further in both introduction and methods (differences in immunostainings, nesting/necroses etc)
2. Which factors were adjusted for in the multivariable models?
Author Response
We thank the reviewer for taking the time to review our manuscript and for the precious comments. By addressing those, we believe that the overall quality of the manuscript is sensibly increased and hope it is now suitable for publication in the Journal.
Thank you for an interesting and well-written paper.
1. Could you please describe the NECs further in both introduction and methods (differences in immunostainings, nesting/necroses etc)
We thank the reviewer for this suggestion. NEC are mainly distinguished from NET based on morphology features, while IHC staining are used to determine primary site (e.g. TTF-1 for lung, CDX2 for the gastrointestinal tract, and so on). We added description of NEC pathology features (and NET for comparison) in the Introduction (lines 46-53) and NEC exclusion criteria in the “2.1. Patient selection” paragraph in the Methods section (line 84-85).
2. Which factors were adjusted for in the multivariable models?
We thank the reviewer for this question. The multivariable model was fitted with all the variables that were significant in the univariate analysis at the p-value <0.1 level. Below is reported Table S1 for your reference in which type of chemotherapy and Ki67 have been included as covariates in the multivariate model. This has been clarified in the “2.3. Statistical analysis” paragraph of the Methods section, line 127.

Round 2
Reviewer 1 Report
happy with the authors response. While i do see little benifit of this study to a wider audience without more detailed summary statistics and more detailed analytics (which I believe should not be an issue with local authorities if addressed appropriatelly), i believe this is still publishable since pushing the boundaries of theuraputics in this cancer area is important.
ok with english.